# Green Synthesis of Free Standing Cellulose/Graphene Oxide/Polyaniline Aerogel Electrode for High-Performance Flexible All-Solid-State Supercapacitors

**DOI:** 10.3390/nano10081546

**Published:** 2020-08-07

**Authors:** Yueqin Li, Zongbiao Xia, Qiang Gong, Xiaohui Liu, Yong Yang, Chen Chen, Changhao Qian

**Affiliations:** 1Co-Innovation Center of Efficient Processing and Utilization of Forest Resources, Nanjing Forestry University, Nanjing 210037, China; zongbiaoxia@163.com (Z.X.); qianggong@163.com (Q.G.); xiaohuiliu@163.com (X.L.); yongyang06@163.com (Y.Y.); 2College of Chemical Engineering, Jiangsu Key Lab for the Chemistry and Utilization of Agricultural and Forest Biomass, Nanjing Forestry University, Nanjing 210037, China; chenchen@163.com (C.C.); changhaoqian@163.com (C.Q.)

**Keywords:** cellulose, graphene oxide, polyaniline, aerogel electrode, supercapacitors

## Abstract

The cellulose/graphene oxide (GO) networks as the scaffold of free-standing aerogel electrodes are developed by using lithium bromide aqueous solution, as the solvent, to ensure the complete dissolution of cotton linter pulp and well dispersion/reduction of GO nanosheets. Polyaniline (PANI) nanoclusters are then coated onto cellulose/GO networks via in-situ polymerization of aniline monomers. By optimized weight ratio of GO and PANI, the ternary cellulose/GO_3.5_/PANI aerogel film exhibits well-defined three-dimensional porous structures and high conductivity of 1.15 S/cm, which contributes to its high areal specific capacitance of 1218 mF/cm^2^ at the current density of 1.0 mA/cm^2^. Utilizing this cellulose/GO_3.5_/PANI aerogel film as electrodes in a symmetric configuration supercapacitor can result in an outstanding energy density as high as 258.2 µWh/cm^2^ at a power density of 1201.4 µW/cm^2^. Moreover, the device can maintain nearly constant capacitance under different bending deformations, suggesting its promising applications in flexible electronics.

## 1. Introduction

The increasing development of wearable electronic devices has triggered an urgent demand for the adaptive power system to provide sufficient energy for long-term operations. Flexible supercapacitors (SCs), emerging as a promising power source of high-performance electronic devices, have aroused tremendous interest due to their high-power density that can fill the gap between rechargeable batteries and conventional capacitors [1,2]. To obtain high performance for flexible SCs, various flexible electrodes have been constructed by traditional metal oxides, nano-carbons, conductive polymers and their composites [3], as well as new materials comprising two-dimensional Mxenes, polyoxometalates (POMs), metal-organic frameworks (MOFs), etc. [4]. Nevertheless, several factors, including the high cost of raw materials, complex preparing process, and hard mass production are still challenging for practical applications. On the other hand, the appetite for sustainable and biodegradable materials in electronics has been significantly increased as the global warming and energy crisis continue to increase. Recent studies have focused on developing highly flexible, free-standing, and binder-free electrodes using cost-effective and renewable raw polymeric materials [5,6,7]. In this scenario, commercial cellulose products, such as A4 paper, air-laid paper, and Kimwipes, as well as various kinds of nanocellulose (e.g., cellulose nanofibrils (CNFs), cellulose nanocrystals (CNCs), and cellulose nanoyarn (CNY)), have been frequently used as sustainable, flexible and foldable scaffolds as structural matrixes for the conducting fillers in supercapacitors [8,9,10,11].

Carbon-based materials (porous carbon, carbon nanotubes (CNTs), graphene, Mexene, etc.) and conducting polymers (polypyrrole (PPy), polyaniline (PANI), poly(3,4-ethylenedioxythiophene): poly(styrene sulfonate) (PEDOT:PSS)) are the most studied and promising electrode materials [12,13]. Nevertheless, the prior researches have proved that the single-component of the carbon-based materials or conducting polymers as electrode materials were not satisfied enough to meet the real application due to the substandard energy density in supercapacitors. Thus, the rational design of hierarchical and porous hybrids is still important for the improvement of electrochemical performances of supercapacitors. In general, cellulose-based substrates are combined with conductive polymers, including polypyrrole (PPy) and PANI, by in-situ polymerization in monomer solutions [14,15,16]. The addition of the extra CNTs or a graphene component makes the preparation more complex, mainly through the “vacuum filtration” strategy or the in-situ culture on bacterial cellulose (BC) [17,18,19,20]. For example, Xu et al. fabricated the PANI/graphene/BC film for supercapacitor electrodes via pouring PANI/GO nanocomposites into BC paper under vacuum [18]. BC/graphene/PANI electrodes were produced by in-situ culture and polymerization [19]. In another study, Dong et al. deposited CNTs on cellulose paper by a “dipping and drying method”, followed by in-situ chemical polymerization of aniline monomers, to get stacked up layers in composite networks [21]. Nevertheless, the filtration of CNTs or graphene into the inner core of cellulose paper could not ensure uniform distribution of conducting fillers in the polymeric matrix. In addition, dip-coating in the solution of CNTs or graphene is challenging due to the limited loading of active materials on the film-like sample, resulting in a small areal capacitance of the whole electrodes.

Very recently, aerogel-based electrodes have shown promising applications in supercapacitors, since their high specific surface area and well inter-connected three-dimensional (3D) texture can load a high content of active materials [22,23,24]. The cellulose/carbon or cellulose/conducting polymers aerogels can be facilely prepared by the solvent mixing method [25,26,27], and the doping content and 3D microporosity can be tunable during the gelification process [28,29]. For instance, a 3D CNFs/PANI aerogel electrode has been prepared by the supramolecular self-assembly method by mixing of CNFs and PANI nanocomposite suspension. Unfortunately, the maximum specific capacitance of CNF/PANI aerogel was only of 59.26 mF/cm^2^ at 10 mV/s [30]. CNFs/rGO/PPy aerogel electrodes were reported to have 1.77 mg/cm^2^ of the active materials and exhibited high areal capacitance of 400 mF/cm^2^ at 0.25 mA/cm^2^, with the capacitor retention rate of only 75.6% [31]. Nanocellulose-supported hierarchical-structured polyaniline/nanocarbon nanocomposite aerogel electrodes have been prepared by via layer-by-layer assembly, which achieved high 1.59 F/cm^2^ at a scan rate of 1 mV/s [32]. Even so, the specific capacitance, rate capability, and retention performance of these reported electrodes are still far from satisfactory for real applications in electronics. Further studies are still necessary to develop flexible cellulose-based electrodes with robust mechanical properties and superior electrochemical performance for flexible and wearable supercapacitors.

In this paper, natural and low-cost cotton linter pulp and commercial single-layer GO were used as raw materials. The regeneration of cellulose fibers and uniformly distribution of the GO nanosheets within the cellulose fibers were easily conducted in LiBr aqueous solution at 120 °C. Fortunately, GO was reduced during the solution processing. To make better use of the conductivity and flexibility of polyaniline, aniline was soaked and in-situ polymerized into this porous cellulose/GO hydrogel matrix to form binder-free composite electrodes. Such a green procedure provides a facile and scalable route for fabricating a robust and high porous 3D network of cellulose-based aerogel electrodes. The effects of different contents of GO and PANI on the electrical conductivity and electrochemical properties were investigated. Due to the three-dimensional porous structures, high conductivity, and remarkable wettability of cellulose/GO/PANI aerogel electrodes, the all-solid-state symmetric supercapacitor, assembled with H_2_SO_4_/poly(vinyl alcohol) (PVA) gel as the electrolyte, can deliver an outstanding areal specific capacitance of 1858.7 mF/cm^2^ at 1.0 mA/cm^2^ and high energy density up to 258.2 µWh/cm^2^ at a power density of 1201.4 µW/cm^2^. Besides, the tandem devices can easily light up a red light-emitting diode for several minutes. These results indicate the cellulose/GO_3.5_/PANI aerogel-based supercapacitors have excellent performance.

## 2. Materials and Methods

### 2.1. Materials

Cotton linter pulp (α-cellulose > 95% and D.P. = 720) was provided by Anhui Snow Dragon Fiber Tech. Co. Ltd. (Suzhou, China). Single-layer graphene oxide (GO) powder was obtained from Nanjing XFNANO Materials Tech Co. Ltd. (Nanjing, China) Lithium bromide (LiBr), aniline (ANI), and ammonia persulfate (APS) were purchased from Aladdin Chemical Co. (Shanghai, China). Other chemicals and reagents of analytical grade were purchased from Sinopharm Group Chemical Reagent Co. Ltd. (Shanghai, China) and used as received.

### 2.2. Preparation of PANI/GO/Cellulose Composites

Briefly, GO powder was added into a LiBr aqueous solution (60 wt.%, 14 mL) and ultrasonicated for half an hour to gain a uniform dispersion. Then, cotton linter pulp (0.25 g) was added and heated up to 120 °C with vigorous stirring, until the cellulose was completely dissolved. After that, the mixture was allowed to gradually cool to become a homogeneous gelatinous form. The obtained cellulose/GO hydrogel blocks were immersed into the deionized water for about 2 days, and then cut into discs with 2 mm thickness and 17.15 mm diameter. Various amount of GO powder was used to keep GO content of 1.0, 2.0, 3.5, and 5 wt.% in the hydrogels, as denoted as cellulose/GO_1.0_, cellulose/GO_2.0_, cellulose/GO_3.5_, and cellulose/GO_5.0_, respectively. The pure cellulose hydrogel without GO was prepared for comparison.

To load the PANI conducting fillers, the hydrogel samples were soaked in an ANI/HCl solution (1 M ANI dissolved in 1 M HCl) to ensure aniline molecules were fully dispersed in the pore of the hydrogel matrix. Subsequently, 10 mL of APS/HCl acid solution (1 M/1 M) was added and gently stirred continuously in an ice bath for 10 h. Finally, the samples were taken out and rinsed with deionized water several times. Finally, the cellulose/PANI and cellulose/GO/PANI composites were obtained after freeze-drying.

### 2.3. Fabrication of All-Solid-State Supercapacitors

The PVA/H_2_SO_4_ gel electrolyte was prepared according to our previous work [33]. Then, the PVA/H_2_SO_4_ gel was pasted onto one side of aerogel electrodes; every two pieces were pressed together under 10 MPa for 10 min to make sure the gel electrolyte was well combined. On the other side of the electrode, a silver paste was coated and the aluminum foil was connected. The working area was 2.31 cm^2^ and the thickness of the device was 0.1 cm.

### 2.4. Characterization of As-Prepared Materials

Fourier Transform Infrared spectrum (FT-IR) was recorded on a Perkin-Elmer Spectrum One spectrometer (Perkin-Elmer, Shanghai, China) in a range of 4000–400 cm^−1^ using KBr pellets. The morphology observation of the samples was examined by a Hitachi SU-8010 field emission scanning electron microscope (Hitachi, Tokyo, Japan). Thermogravimetric analysis (TGA) was conducted on a NETZSCH STA 449 F3 instrument (NETZSCH, Shanghai, China), and the samples were heated up to 800 °C at a heating rate of 10 °C/min under nitrogen flow of 5 mL/min. X-ray photoelectron spectroscopy (XPS) analyses were carried out on an AXIS UltraKratos photoelectron spectrometer (SHIMADZU, Kyoto, Japan), utilizing a monochromatic 150 W Al X-ray source. Nitrogen adsorption isotherms were collected at 77 K using an autosorb iQ instrument (Quantachrome, Boynton Beach, FL, USA) volumetric gas adsorption analyzer. The specific surface area and the pore size distribution of the samples were calculated according to the Brunauer–Emmett–Teller (BET) and Barrett–Joyner–Halenda (BJH) methods, respectively. The electrical conductivity of aerogel samples was measured using the previously reported method [34,35]. Typically, the as-prepared samples were cut into 1 cm × 1 cm, and pressed at 10 MPa for 10 s. The thickness of the samples was estimated by using a micrometer. Bulk resistance (*R*) of the samples was measured at room temperature using a digital multimeter. The conductivity, σ was obtained from the equation: σ = *L*/(*R* × *W* × *d*), where *L*, *W*, and *d* refers to the length, width, and thickness in cm, respectively.

The electrochemical characterizations were carried out on a CHI760E electrochemical workstation (CHI instrument, Shanghai, China). In a 3-electrode configuration, samples with 1 × 1 × 0.1 cm^3^ dimension, saturated calomel electrode and platinum plate are working, reference, and counter electrodes, respectively; 1 M H_2_SO_4_ aqueous solution was used as the electrolyte. Specifically, electrochemical impedance spectroscopy (EIS) spectra were conducted over a frequency range of 10 MHz to 100 kHz by with an AC sinusoid signal of 5 mV amplitude. To calculate the areal specific capacitance (*C*_s_ in mF cm^−2^) and Coulombic efficiency (η), the following equations were used [36,37]:(1)$$C_{s}=\frac{I\times t}{S\;\times \Delta V}$$
(2)$$\eta \;=\frac{t_{D}}{t_{C}}\times 100\%$$
where *I* is the charge–discharge current (mA), *t* represents the discharge time (s), S is the area of the electrode (cm^2^), and Δ*V* is the potential window of the discharge curve excluding initial drop (V). For the symmetric supercapacitor device, the areal specific capacitance (*C*_a_ in mF cm^−2^), the energy density (*E*_a_ in µW h cm^−2^) and the power density (*P*_a_ in µW cm^−2^) were calculated according to the following equations [36,37,38,39]:(3)$$C_{a}=\frac{2I\times t}{S\;\times \Delta V}$$
(4)$$E_{a}=\frac{0.5C_{a}\Delta V^{2}}{3.6}$$
(5)$$P_{a}=\frac{3600E_{a}}{t}$$

## 3. Results and Discussion

### 3.1. Preparation, Morphology, and Structure of the Ternary Cellulose/GO/PANI Composite

As illustrated in Figure 1, the cellulose/GO/PANI composite was prepared in three steps, including dissolving cotton linter pulp and dispersing GO simultaneously into the aqueous solution of LiBr, adsorption of aniline monomer onto the network, and in-situ polymerization of the PANI component. The regeneration of the cellulose/GO hydrogel was a black block, and then it was cut into thin sheets with a thickness of roughly 2.0 mm. The black sheets turned dark green after the growth of PANI in the 3D networks. Due to the coating of hydrophobic PANI on the surface of the sheets, wettability was evaluated by distilled water contact angles method. As shown in the bottom of Figure 1, a water droplet penetrates into the cellulose/GO_3.5_/PANI sample instantly. This phenomenon could be due to the sufficient porosity on the smooth surface of the sheet. Although the contact angle was undetectable, the super wettability of this cellulose-based surface will promote the electrolyte receptivity that is favorable for the electrochemical performance.

The chemical composition of the obtained samples was characterized by FT-IR spectroscopy. As presented in Figure 2a, the regenerated cellulose and GO samples exhibit several absorption peaks in good agreement with those reported elsewhere [33,40]. When doped with a small amount of GO with cellulose, the cellulose/GO composite exhibits all the characteristic peaks corresponding to cellulose and GO, except that the characteristic peak of carbonyls (C=O) at 1730 cm^−1^ decreased dramatically, indicating the reduction of the oxygen functional groups [41].The neat PANI showed the absorption peaks at 1575, 1490, and 1132 cm^−1^, assignable to the characteristic vibrations of the benzene ring, the peak at 1297 and 799 cm^−1^ corresponding to the C–N stretching and C–H out-of-plane bending vibrations, respectively. However, the peaks located at 1575, 1297, and 1132 cm^−1^ of PANI show an obvious blue shift to 1588, 1312, and 1153 cm^−1^ in the spectrum of the cellulose/GO_3.5_/PANI sample, suggesting the π–π interaction exists between the GO and conjugated structure of the PANI, which would strengthen the electrical conductivity of the materials [42,43]. Additionally, the absorption peak at 3430 cm^−1^ due to the stretching vibration of OH groups in cellulose/GO_3.5_ sample is down-shifted to 3375 cm^−1^ in cellulose/GO_3.5_/PANI sample, which indicates the existence of strong hydrogen bonding interaction among the three components [44].

Raman spectroscopy was used to reflect the structural details of the ternary composite. Raman spectra of GO, PANI, cellulose/GO_3.5_, and cellulose/GO_3.5_/PANI composites are demonstrated in Figure 2b. Typically, GO exhibited the characteristic D band at 1350 cm^−1^ and G band at 1586 cm^−1^. Nevertheless, with the thermal blending process in the LiBr solution, the D band of the cellulose/GO_3.5_ showed a noticeable downshift to 1321 cm^−1^, along with an obvious increase in I_D_/I_G_ of 1.47 from original 0.973. These observations suggested the thermal reduction of GO and the existence of highly disordered structures in the as-prepared cellulose/GO_3.5_ sample [45]. Further, after cooperation with PANI, the cellulose/GO_3.5_/PANI composite showed mainly typical bands around 1338 and 1586 cm^−1^, which is attributable to the C–N/C=N and C–C bonds, respectively. These broad bands should have overlaid characteristic peaks of GO and PANI, indicating the complete wrapping of GO nanosheets and cellulose nanofibers by PANI [19].

X-ray diffraction of neat cellulose, GO, PANI, cellulose/GO_3.5_, and cellulose/GO_3.5_/PANI composites are shown in Figure 2c. GO mainly displayed an intense characteristic peak at 11° corresponding to the (001) reflection peak. The cellulose/GO_3.5_ sample exhibited similar X-ray diffraction (XRD) spectrum to that of the pure cellulose, in which diffraction peaks appeared at 2θ = 16.5°, 22.9°, and 34.8° [46], indicating the crystalline type of cellulose is not altered after being doped with GO and the following regeneration. The absence of the characteristic peak at 11° is due to the low content and the thermal reduction as mentioned in Raman analysis. The pure PANI exhibited four distinct peaks centered around 8.9°, 15.2°, 20°, and 25.3°, which is assignable to (001), (011), (020), and (200) crystal planes of PANI in its emeraldine salt form [47]. In the XRD pattern for cellulose/GO_3.5_/PANI composite, not only was the characteristic diffraction peak of cellulose observed at 22.9°, but also low-intensity diffraction peaks of PANI at 15.2° and 25.3°, which indicates the presence of PANI in the ternary composites. However, the intensity of the diffraction peaks of the cellulose/GO_3.5_/PANI composite is apparently decreased. It turns out again that the strong interactions among these components have limited the re-crystallization of the composite aerogel [48].

The chemical bonding in the cellulose/GO_3.5_/PANI composite was further analyzed by XPS measurements. The XPS survey spectrum of GO, PANI, and the cellulose/GO_3.5_/PANI composite are presented in Figure 3a. Typically, except for C and O elements, extra elements of nitrogen, chloride, and sulfur are noted in the cellulose/GO_3.5_/PANI composite. It is noteworthy that the peak related to the element sulfur is due to the residue from a trace of APS. The deconvolution of C 1s spectra of PANI and the cellulose/GO_3.5_/PANI composite are presented in Figure 3b. The major peak line of PANI can be decomposed into four peak lines: 284.6 eV (C–C/C=C, 33.72%), 285.6 eV (C–N, 38.44%), 286.6 eV (C=N/C=N^+^, 22.31%), and 288.1 eV (C=O, 5.53%) [49]. The C=O functional groups present in PANI might be assigned to the formation of benzoquinone and hydroquinone [50] as products of the partial surface oxidation during handling. While the C 1s core-level spectrum of the cellulose/GO_3.5_/PANI sample can be curve-fitted into five subpeaks (Figure 3c): 284.8 eV (C–C/C=C, 21.63%), 285.8 eV (C–N, 24.99%), 286.6 (C=N/C=N^+^, 10.70%), 287.2 (C–O, 22.27%), and 288.1 (C=O, 20.42%) (Figure 3b). The relatively higher ratio of C=N/C=N^+^ to C–N bonds in the ternary composite (43%) is found much lower than that for pure PANI (58%), which could be related to the interactions of the amine groups of the PANI with oxygen functional groups of GO [49]. This observation further supports the presence of strong interactions in line with the FT-IR and XRD characterizations. In addition, the N 1s XPS spectrum of the pure PANI sample is deconvoluted into four peaks at 398.3 eV (=N–, 2.21%), 399.8 eV (–NH–, 75.69%), 401.4 eV (N^+^, 16.10%), and 403.0 eV (–NH^+^, 6.00%). All four types of amine structure can be found in the N 1s spectrum of the cellulose/GO/PANI composite with different contents of 2.12%, 68.18%, 20.64%, and 9.10%, respectively. Specifically, the sum of positively charged nitrogen in the composite is higher than that in pure PANI, indicating a relatively higher doping level of PANI in the composite. The increased proportion of positively charged nitrogen could be ascribed to the interactions between PANI and GO that leads to the restructuring of the benzenoid amine structure of PANI [51], which is also in accordance with the result of the C 1s spectra analysis. Additionally, the high doping level of PANI in this composite is expected to enhance its pseudocapacitive performance as electrode materials [52].

The thermal properties and the content of the PANI (in weight ratio) in the cellulose/GO_3.5_/PANI sample were determined by thermogravimetrical analysis, with the regenerated cellulose, GO, PANI, as well as the cellulose/GO_3.5_ composite used as references. As presented in Figure 3d, a weight loss of ~5 wt.% for all samples up to a temperature of 100 °C is due to the evaporation of physisorbed water. The cellulose aerogel was thermally stable up to 210 °C; a major weight loss commenced from 210 to 450 °C could be attributed to the pyrolysis of the molecular fragment, such as –OH and –CH_2_–OH, and finally decomposed of the main chains. As incorporated with a small amount of GO, the cellulose/GO_3.5_ composite exhibits a slightly higher mass loss than that of neat cellulose, which is associated with the removal of labile oxygen functional groups of GO [53]. Pure PANI is known to degrade in a three-stage weight loss profile, where the weight loss step at 100–300 °C is attributed to the loss of the dopant ions (weight loss of 6%), and the final step (>400 °C) with the weight loss of 36% corresponds to the degradation of backbone chains of the polymer. Interestingly, the TGA profiles of the ternary composite exhibit similar curve features of PANI. With the combination of PANI, the ternary composite is relatively more thermal stable; the degradation rate is much slower than that of cellulose/GO_3.5_ aerogel. The gradual loss in the region at 400–650 °C is estimated to be ca. 15 wt.%, which is easily attributable to the weight fraction of PANI in the ternary composite of cellulose/GO_3.5_/PANI. Additionally, as can be seen in DTG curves, *T*_max_ of PANI was found at 512 °C; however, the corresponding peak in the cellulose/GO_3.5_/PANI composite was up-shifted to 530 °C. The phenomenon could be related to the establishment of hydrogen bonds of PANI with the cellulose and strong π–π interaction with GO components, resulting in the formation of mechanical stable nanocomposites.

The digital image in Figure 4a shows free-standing films of the prepared cellulose, cellulose/GO_3.5_, and cellulose/GO_3.5_/PANI samples. PANI could be seen throughout the cellulose/GO_3.5_/PANI sample with relatively low reflectivity, suggesting uniformity in monomer infiltration and growth. The ternary cellulose/GO_3.5_/PANI sample has very good flexibility, showing to be bendable in Figure 4b. The inner morphological details of the pure cellulose, cellulose/GO_3.5_, and the cellulose/GO_3.5_/PANI composite were characterized by field emission scanning electron microscopes (FE-SEM). The cross-section view of the regenerated cellulose sample shows a network-like and 3D porous structure (Figure 4c–e), and the surface of cellulose nanowhiskers is relatively neat at high magnification. Apparent changes were noticed in the morphology of the cellulose/GO_3.5_ aerogel (Figure 4f–h), in which GO sheets partly fill the micropores to form a smooth wall for macropores. In detail, some smaller GO sheets also attach to the cellulose fiber-like nodes. After the in-situ polymerization of ANI with HCl as dopant, the porous and spongy structure has persevered. PANI, in the form of nanoclusters, grew on the network of cellulose/GO_3.5_, forming loose beaded chains with many interspaces remaining. It is believed that this composite should have a higher specific surface area than bulk materials to facilitate the effective access of the electrolyte ions [54]. Given the above chemical structure and morphology analysis, despite the hydrophobic character of PANI and the hydrophilic character of cellulose, the ternary composites have shown good miscibility and numerous electro-active clusters have been uniformly loaded in the 3D structure.

The gas adsorption–desorption isotherms for the as-prepared cellulose, cellulose/GO_3.5_, cellulose/PANI, and cellulose/GO_3.5_/PANI aerogel samples were carried out by the Brunauer–Emmett–Teller (BET) technique (Figure 5). All samples show type II adsorption isotherms, indicating the presence of many macropores and mesopores. BET and BJH analyses (Table 1) revealed that the regenerated cellulose sample had a well-defined mesopore of 10.9 nm, and high-specific surface area of 137.6 m^2^/g and a total pore volume of 0.38 cm^3^/g. While doped with GO, the surface area and pore feature showed a slight increase, which is resulted from the fact that the presence of graphene sheets can reinforce cellulose nanofibers to reduce shrinkage of aerogel during freeze-drying. Due to the deposition of PANI on cellulose nanofibers, the measured surface area for the cellulose/PANI is much lower than that of the regenerated cellulose, indicating the mesoporous channels could have been partially blocked. Likewise, the cellulose/GO_3.5_/PANI composite shows a surface area of 66.7 m^2^/g, which is 55% lower than that of the cellulose/GO_3.5_ sample. The pore size distribution of the cellulose/GO_3.5_/PANI sample presented a wide distribution of pores from micro to meso ranges, with the average pore size of 22.5 nm, which is the highest one among the tested samples. It is believed that the adequate pore size and large total pore volume of cellulose/GO_3.5_/PANI composites would be beneficial to electrochemical performances, because they provide fast ion transportation pathways through the nanopores of electrode materials during the electrochemical tests [55].

### 3.2. Conductivity and Electrochemical Properties of Cellulose/GO/PANI Composite Electrodes

Conductivity measurements on the various cellulose/GO_1.0_/PANI, cellulose/GO_2.0_/PANI, cellulose/GO_3.5_/PANI, and cellulose/GO_5.0_/PANI samples indicated an average electrical conductivity of 0.50, 0.82, 1.15, and 0.35 S/cm, respectively. The obtained high overall conductivity of the cellulose/GO_3.5_/PANI sample ensures the use as a wire to light up a LED lamp (Appendix A), even at bending and foldable deformations. Further, the cyclic voltammogram (CV) tests for these samples were run to explore the effect of the GO content on the electrochemical performance of the ternary composite electrodes, as shown in Appendix A. As the GO content increased, the area surrounded by the CV curves was not enlarged tremendously. Based on the conductivity and CV performance, the ternary sample with GO content of 3.5 wt.% has shown promising electrical properties among the ternary composites. As anticipated, the cellulose/GO_3.5_/PANI electrode exhibits an enhanced remarkable electrical response when compared with cellulose/GO_3.5_ and cellulose/PANI electrodes (Figure 6a), which can be attributed to the synergistic effect of graphene and PANI as conducting fillers [6]. Meanwhile, the redox current density of this sample becomes larger and larger as the scan rate increased, indicating a good rate capability of the electrode. Two pairs of redox peaks of PANI can be observed at a low rate; however, they vanished at high scan rates, which are attributed to the existence of internal active sites that may not completely carry out redox transition, resulting in the variation of volt-ampere current at high scanning rate.

The charge storage properties were further investigated by the galvanostatic charge–discharge (GCD) measurements. The GCD profiles of the cellulose/GO_3.5_/PANI electrode at various current densities show nonlinear shape with two voltage stages in the discharge process (Figure 6c). At relative lower current densities of 1.0 mA/cm^2^, the former stage (0.8–0.45 V) with relatively short discharge time is ascribed to the electrochemical double layer capacitor (EDLC), while the later stage (0.45–0 V) with much longer discharge time is associated with a combination of EDLC and pseudocapacitive capacitance. A typical comparison of GCD curves of the cellulose/PANI and cellulose/GO_3.5_/PANI composite electrode is presented in Figure 6d. Benefiting from the presence of GO, the potential drop at the beginning of the discharge curve (called “IR drop”) of the cellulose/GO_3.5_/PANI composite is much lower than that of the cellulose/PANI electrode, revealing lower internal resistance of cellulose/GO_3.5_/PANI composite than cellulose/PANI composite. According to Equation (1), the areal specific capacitance of cellulose/GO_3.5_/PANI composite is found to be 1218 mF/cm^2^ at the current density of 1.0 mA/cm^2^. Similar calculations were done for the cellulose/PANI electrode; however, much lower areal specific capacitance was found (Figure 6e), thus confirming the high loading capacity of the cellulose/GO_3.5_/PANI composite. Actually, to evaluate the contribution of PANI to energy storage, a series of ternary cellulose/GO/PANI samples (GO of 3.5 wt.%) were prepared from different aniline concentrations of 0.5, 0.75, 1.0, and 1.25 M. Their electrochemical performances along with comparison with those of the cellulose/GO_3.5_/PANI electrode are presented in Appendix A. The largest enclosed area and the longest charging/discharging time of the cellulose/GO_3.5_/PANI electrode indicates the best capacitive performance among the four samples. Ideal performance on the cellulose/GO_3.5_/PANI electrode may be due to its unique pore structure and the appropriate size of PANI nanoclusters. When the aniline concentration increased to 1.25 M, the thicker coating layer of PANI might cause its underutilization that induces a decrease of the specific capacitance [56].

Besides, the EIS measurements of the cellulose/PANI and cellulose/GO_3.5_/PANI electrodes were examined to gain more insight into the interfacial transfer kinetics of electrolyte ions. As presented in Appendix A, the Nyquist plots show a single semicircle at high-frequency region with the *x* intercept of 1.6 and 2.4 Ω for cellulose/PANI and cellulose/GO_3.5_/PANI, respectively, reflecting small equivalent series resistance. The diameter of the semicircles corresponding to the charge transfer resistance is obtained by a fitting circuit, in which much smaller *R*_ct_ is found for the cellulose/GO_3.5_/PANI (3.1 Ω) than that of the cellulose/PANI (12.3 Ω), implying a much lower charge-transfer resistance. The result is consistent with the improved conductivity of the cellulose/GO_3.5_/PANI compared to that of the cellulose/PANI electrode. Figure 6f shows the electrochemical stability of the cellulose/GO_3.5_/PANI electrode during continuous charging and discharging at a constant current density of 15 mA/cm^2^. The specific capacitance can preserve 83% after 1000 charge–discharge cycles. The Coulombic efficiency of ~100% over 1000 cycles is indicative of outstanding electrochemical reversibility. The proper degradation of capacitance is likely attributed to the structural deterioration of PANI and volumetric changes from expansion and contraction of the electrode, which is still a tough problem for PANI-based supercapacitors [57]. What is more, the cellulose/GO_3.5_/PANI electrode has shown comparability or even superiority in performance when compared with other reported composite materials. Some of the details on enhancements in the areal specific capacitance and the cyclic stability are summarized in Appendix A.

### 3.3. Electrochemical Performanc of the Fabricated All-Solid-State Supercapacitors

To evaluate the potential practical application, we assembled all-solid-state supercapacitors with a symmetrical sandwich structure based on cellulose/GO_3.5_/PANI composite electrodes using PVA/H_2_SO_4_ gel as the electrolyte. Typical CV profiles of a single device in the range −0.2 to 0.8 V at different scan rates are shown in Figure 7a. The asymmetric triangular shape of GCD curves also indicates the pseudo-capacitive properties of the cellulose/GO_3.5_/PANI electrode, as shown in Figure 7b. The areal capacitances calculated from GCD curves drop from 1858.7 to 909.4 mF/cm^2^ as the current density increases from 1.0 to 8.0 mA/cm^2^ (Figure 7c). Since the thickness of the device is of ~0.1 cm, the maximum volumetric capacitance is determined to be 18.5 F/cm^3^ at 10.0 mA/cm^3^. The maximum capacitive performance of our device is substantially higher than that of previously reported supercapacitors based on the CNF/rGO/CNT hybrid aerogel electrode (216 mF/cm^2^ at 0.5 A/g) [58], PANI/Ag/CNF aerogel (176 mF/cm^2^ at the scan rate of 10 mV/s) [59], polypyrrole/rGO/BC flexible electrode (790 mF/cm^2^ at 1.0 mA/cm^2^) [60], and PANI/BC/graphene paper (1.93 F/cm^2^ at 0.25 mA/cm^2^) [61]. The high areal capacitance and rate capability are ascribed to the high compatibility of the cellulose-based electrode with gel electrolyte, which endows effective ion diffusion pathways to the interior of the PANI coating [31].

Moreover, according to Equations (4) and (5), the energy and power density of the assembled supercapacitor is obtained from the GCD curves and is manifested in the Ragone diagram (Figure 7d). The device achieves a maximum energy density of 258.2 µWh/cm^2^ at a power density of 1201.4 µW/cm^2^, and still can store an energy density of 126.3 µWh/cm^2^ at the power density of 15.1 mW/cm^2^ as the current density increased from 1.0 to 8.0 mA/cm^2^. The corresponding areal energy and power density values are much higher than the recently reported supercapacitors based on PANI/CNT paper (29.4 µWh/cm^2^ at 391 µW/cm^2^) [21], CNFs/rGO/PPy (60.5 µWh/cm^2^ at 0.1 MW/cm^2^) [31], CNF/RGO/CNT aerogel (28.4 µWh/cm^2^ at 9.5 MW/cm^2^) [58], RGO/MnO_2_ paper (35.1 µWh/cm^2^ at 37.5 µW/cm^2^) [62], rGO/PANI-PSS papers (40.7 µWh/cm^2^ at 500 µW/cm^2^) [63], PANI/PVA-H_2_SO_4_ chemical hydrogel film (42 µWh/cm^2^ at 160 µW/cm^2^) [64], CNT@PANI film (50.98 µWh/cm^2^ at 2294 µW/cm^2^) [65], CNT/activated carbon fiber felt (112 µWh/cm^2^ at 490 µW/cm^2^) [66], PANI/GN/BC paper (120 µWh/cm^2^ at 100 µW/cm^2^) [67], and carbon microfiber/MWCNT (9.8 µWh/cm^2^ at 189.4 µW/cm^2^) [68]. EIS analysis of the single device (Figure 7e) indicates that the *R*_s_ and *R*_ct_ are of 3.2 and 1.9 Ω, respectively. The electrochemical properties of the fabricated supercapacitor were also tested at various bending angles (from 0° to 180°, Figure 7f). The overlapped CV curves demonstrate the device has excellent electrochemical behavior under various bending deformations. The relatively small *R*_s_ and *R*_ct_ and excellent flexibility indicate its appropriate configuration design, and some larger area supercapacitors could be fabricated for integrating into wearable electronics.

To access the real-life applications of the as-assembled supercapacitor, more devices can be connected in series and parallel connections. For instance, Figure 8a shows the CV curves of two devices in series and parallel connections along with the comparison of the single device. The observations indicate an extended the operating window of two devices in series and an enlarged capacity by parallel under the same current density. Moreover, three times the operating voltage was obtained by three devices connected in series, which can drive a red LED bulb to light up for almost 4 min, as manifested in Figure 8b,c. All of these results demonstrate that our flexible supercapacitors based on the cellulose/GO_3.5_/PANI electrode are promising in flexible and lightweight electronics.

## 4. Conclusions

Free-standing cellulose/GO/PANI composite aerogels can be delicately synthesized by in-situ polymerization of aniline in the cellulose/GO three-dimensional frameworks. With the high conductivity, the cellulose/GO/PANI aerogels can serve as electrode materials in the absence of any other binders, and retain high flexibility that can be bent to a large degree. Excellent electrochemical performances are achieved for the cellulose/GO_3.5_/PANI electrode, which produced a maximum areal specific capacitance of 1218 mF/cm^2^ at a current density of 1.0 mA/cm^2^ and maintained good cycling stability with capacitance retention of 83% after 1000 charge–discharge cycles. Moreover, benefiting from the optimized weight ratio of GO and PANI, the special three-dimensional porous structure, and the synergistic effect between components, the assembled supercapacitor based on the cellulose/GO_3.5_/PANI electrode can deliver an outstanding energy density as high as 258.2 µWh/cm^2^ at a power density of 1201.4 µW/cm^2^, which is superior to that of the aerogel-based flexible supercapacitors reported previously. Therefore, this study provides a promising and efficient way for the preparation of ideal electrode materials for high-performance flexible energy storage devices.

## Figures and Tables

**Figure 1 nanomaterials-10-01546-f001:**
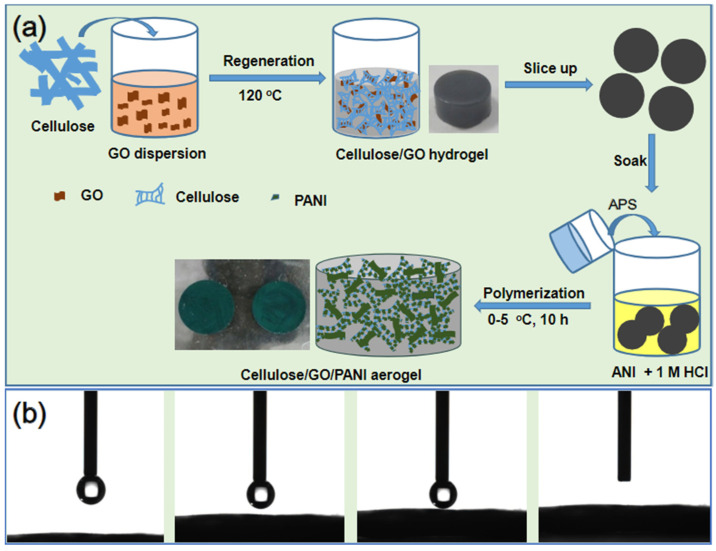
(**a**) Synthetic pathway of the cellulose/ graphene oxide (GO)/ Polyaniline (PANI) composites. (**b**) The water droplet contact process on the surface of the cellulose/GO_3.5_/PANI composite.

**Figure 2 nanomaterials-10-01546-f002:**
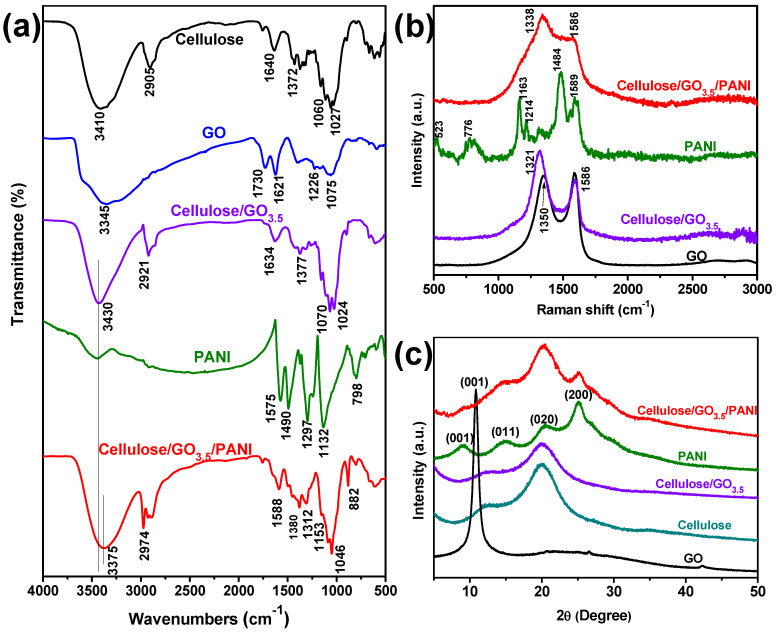
(**a**) Fourier Transform Infrared spectrum (FT-IR), (**b**) Raman, and (**c**) X-ray diffraction (XRD) spectra of neat cellulose, GO, PANI, cellulose/GO_3.5_ and cellulose/GO_3.5_/PANI composites.

**Figure 3 nanomaterials-10-01546-f003:**
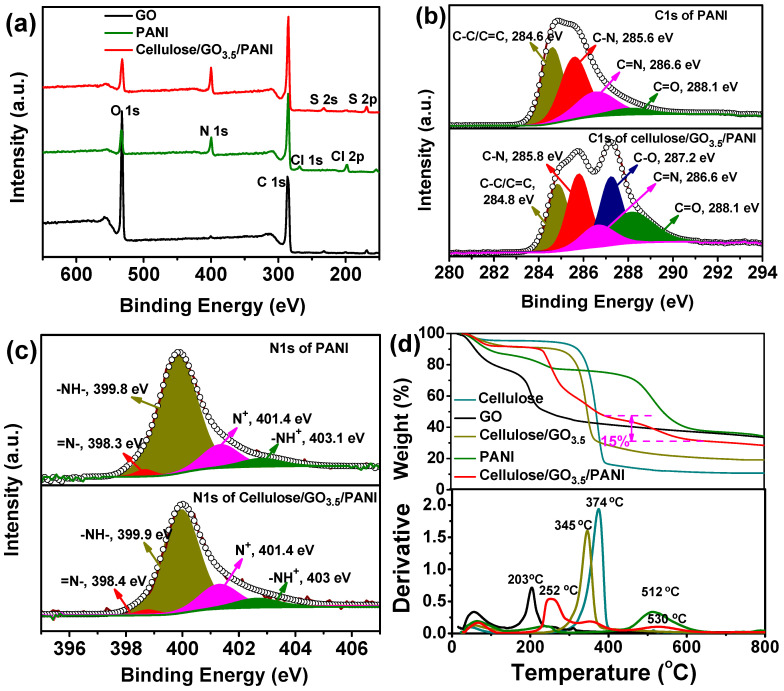
(**a**) X-ray photoelectron spectroscopy (XPS) survey spectra of the pristine GO, neat PANI, and the cellulose/GO_3.5_/PANI composite. (**b**) Comparison of C 1s spectra of PANI and the cellulose/GO_3.5_/PANI composite. (**c**) N 1s spectra of the cellulose/GO_3.5_/PANI composite and pure PANI. (**d**) Thermogravimetric analysis (TGA) pattern of pure cellulose, GO, PANI, cellulose/GO_3.5_, and cellulose/GO_3.5_/PANI samples.

**Figure 4 nanomaterials-10-01546-f004:**
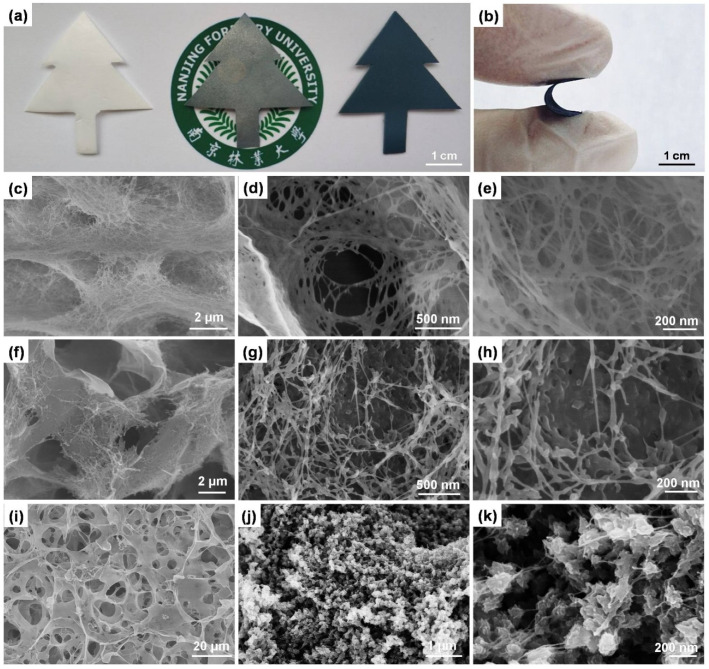
(**a**) Digital image of the pure cellulose (left), cellulose/GO_3.5_ (middle), and cellulose/GO_3.5_/PANI (right) cut in a tree shape. (**b**) Digital photograph of the cellulose/GO_3.5_/PANI under bending. Field emission scanning electron microscopes (FE-SEM); images of pristine cellulose sample (**c**–**e**), cellulose/GO_3.5_ sample (**f**–**h**), and cellulose/GO_3.5_/PANI sample (**i**–**k**) at different magnifications.

**Figure 5 nanomaterials-10-01546-f005:**
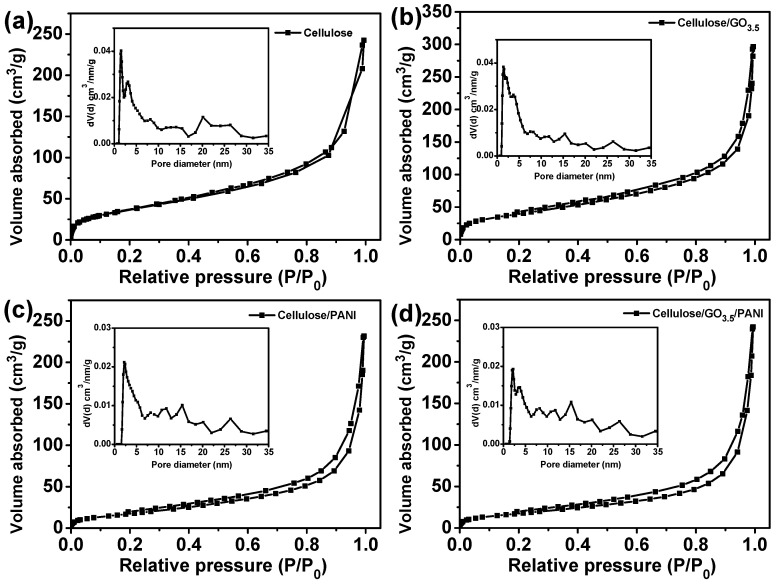
N_2_ sorption isotherms of the regenerated cellulose (**a**), cellulose/GO_3.5_ (**b**), cellulose/PANI (**c**), and cellulose/GO_3.5_/PANI (**d**) samples. Inset is the corresponding pore size distribution.

**Figure 6 nanomaterials-10-01546-f006:**
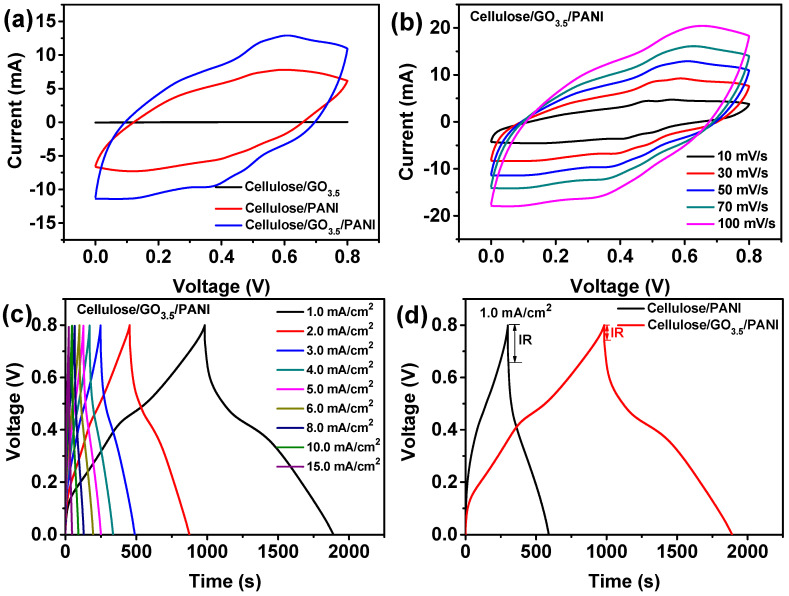
(**a**) Typical cyclic voltammogram (CV) curves of the cellulose/GO_3.5_, cellulose/PANI, and cellulose/GO_3.5_/PANI electrodes at 50 mV/s. (**b**) CV profiles of the cellulose/GO_3.5_/PANI electrode at various sweep rates. (**c**) Galvanostatic charge–discharge (GCD) curves of the cellulose/GO_3.5_/PANI electrode at different current densities. (**d**) Comparative GCD curves of cellulose/PANI and cellulose/GO_3.5_/PANI at a current density of 1.0 mA/cm^2^. (**e**) Plots of areal specific capacitance vs. current density. (**f**) Cyclic stability and Coulombic efficiency (η) of the cellulose/GO_3.5_/PANI electrode at a current density of 15 mA/cm^2^. The inset demonstrates the GCD curve for the last five cycles.

**Figure 7 nanomaterials-10-01546-f007:**
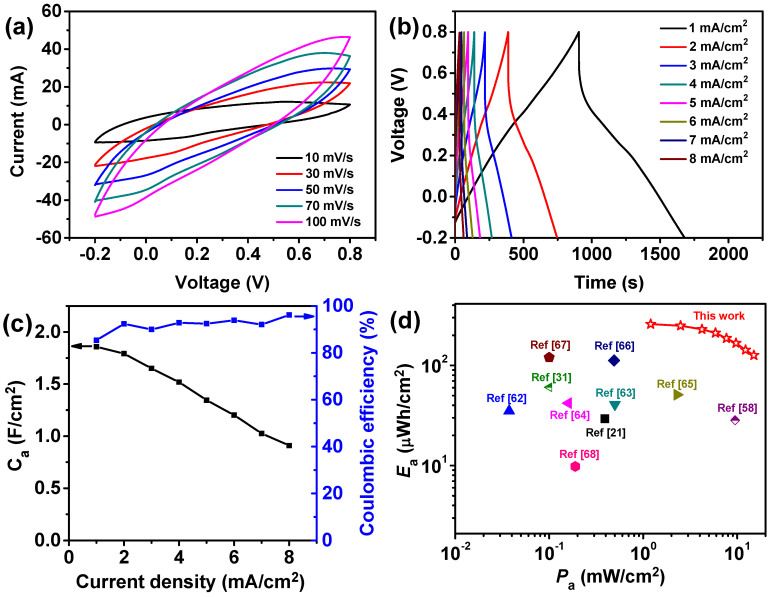
(**a**) Typical CV curves of the assembled supercapacitor at different scan rates. (**b**) GCD curves of the single device at different current densities. (**c**) The areal capacitance varied with different current densities. (**d**) Ragone plots and comparisons with those of the reported solid-state supercapacitors. (**e**) Nyquist plot of the supercapacitor with an equivalent circuit in the inset. (**f**) CV profiles of the device at flat and at 90°, 120°, and 180° bending deformations.

**Figure 8 nanomaterials-10-01546-f008:**
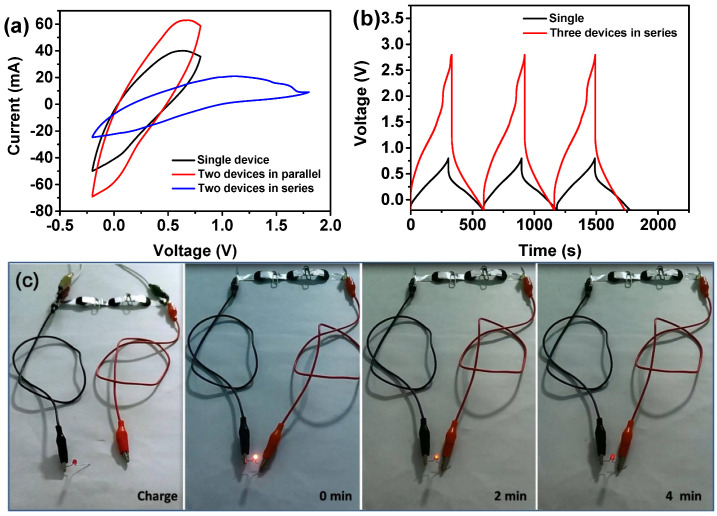
(**a**) CV profiles of the single device, two devices in series, and two devices in parallel connection at a scan rate of 50 mV/s. (**b**) GCD profiles of the single device and three devices in series at a current density of 2 mA/cm^2^. (**c**) Optical pictures of three devices in series connection to light a LED lamp for 4 min.

**Table 1 nanomaterials-10-01546-t001:** Brunauer–Emmett–Teller (BET) analysis of the regenerated cellulose, cellulose/GO_3.5_, cellulose/PANI and cellulose/GO_3.5_/PANI.

Sample	S_BET_	Pore Volume (cm^3^/g)	Pore Size (nm)
cellulose	137.6	0.38	10.9
cellulose/GO_3.5_	147.0	0.46	12.5
cellulose/PANI	68.7	0.36	20.8
cellulose/GO_3.5_/PANI	66.7	0.37	22.5

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
