# Peer review of "Green Synthesis of Free Standing Cellulose/Graphene Oxide/Polyaniline Aerogel Electrode for High-Performance Flexible All-Solid-State Supercapacitors"

_nanomaterials, 2020, doi:10.3390/nano10081546_

Round 1

Reviewer 1 Report

This article describes the Green synthesis of free standing cellulose/graphene oxide/polyaniline aerogel electrode for high performance flexible all-solid-state supercapacitors. The authors claimed that their results indicate the device can maintain nearly constant capacitance under different bending deformations. I recommend several additions that the authors may like to consider before the publication is suitable in your Journal.

  1. Page 1, lot of notation without define, it’s very hard to understand wide readers. e.g. GO, PANI
  2. Page 1 line 46-49 “promising electrode materials.” Relevant reference should be added.
  3. Page 4 line 171 “distilled water contact angles method.” And line 174 was mentioned about undetectable contact angle?? The authors should provide more information about contact angles mtd.
  4. Page 4-5 Figure 1b, “The water droplet contact process on the surface of the cellulose” There is no any information on figure, what does mean of 4 figs, the authors should give clear image or photograph of surface  

Author Response

Page 1, lot of notation without define, it’s very hard to understand wide readers. e.g. GO, PANI

Response: The abbreviations have been defined in Page 1.

Page 1 line 46-49 “promising electrode materials.” Relevant reference should be added.

Response: Relevant references have been added.

Page 4 line 171 “distilled water contact angles method.” And line 174 was mentioned about undetectable contact angle?? The authors should provide more information about contact angles mtd.

Response: Distilled water contact angles method was run by three investigators, the obtained result are reproducible. The aim is to access the wettability of the ternary sample. As shown in Figure 1b, a water droplet penetrates into the cellulose/GO3.5/PANI sample instantly. This phenomenon could be due to the sufficient porosity on the smooth surface of the sheet. Although the contact angle was undetectable, it is satisfied enough to prove the super wettability of this cellulose-based surface, which will promote the electrolyte receptivity that is favorable for the electrochemical performance.

Page 4-5 Figure 1b, “The water droplet contact process on the surface of the cellulose” There is no any information on figure, what does mean of 4 figs, the authors should give clear image or photograph of surface.

Response: In our cases, thick PANI layer is on two side of the cellulose/GO3.5/PANI aerogel. The cellulose/GO3.5/PANI sample shows the instant absorption of the water droplet, indicating the superhydrophilic behavior. The wettability and permeability of the porous structure tends to promote the diffusion of the electrolyte ion within the networks and thus enhancement in electrochemical performance. Similar experiment have been conducted in the reference [1].

  1. Yin, B.-S., Zhang, S.-W., Ke, K., Wang, Z.-B., Advanced deformable all-in-one hydrogel supercapacitor based on conducting polymer: Toward integrated mechanical and capacitive performance, J. Alloys Compd. 2019, 805, 1044-1051.

Reviewer 2 Report

Li et al  have prepared a free standing GO/polyaniline aerogel electrode for supercapacitor applications. I highly appreciate the authors for their effort in this manuscript and for providing all the necessary measurements and discussing the results deeply. I highly recommend the article to published in the Nanomaterials journal without any further modifications.

Author Response

Li et al  have prepared a free standing GO/polyaniline aerogel electrode for supercapacitor applications. I highly appreciate the authors for their effort in this manuscript and for providing all the necessary measurements and discussing the results deeply. I highly recommend the article to published in the Nanomaterials journal without any further modifications.

Response: Thank you very much for your reviewing.

Reviewer 3 Report

MANUSCRIPT NUMBER: nanomaterials-887484

The article describes the fabrication and electrochemical performance of cellulose-based aerogel electrodes to be used in SCs. Although the topic of obtaining this kind of materials in a large scale is of great interest, the paper should be REJECTED for publication in Nanomaterials in its present form. The following issues must to be addressed in depth:

  1. English should be carefully reviewed
  2. In the introduction section authors compare the electrochemical performance of different aerogel-based electrode materials previously reported. However the units of the specific capacitance values are different (F/g vs. mF/cm2) in the papers selected. This issue should be considered in order to make a suitable comparative.
  3. About contact angle measurements: how do the authors explain the results obtained? Why it has not been possible to obtain these values? What happened with the other composites?
  4. FTIR experiments: authors should replace “chemical structures” by “chemical composition”. Furthermore, how do the FTIR data agree with the previously stated GO reduction during the synthesis process? The authors should be clearly clarifying this point. If the GO is reduced during the synthesis process, why it is not reflected in the corresponding spectra?
  5. Authors state that the obtained samples showed “isotherms type IV”. Where is the hysteresis loop?
  6. Electrochemical characterization:

- Authors should improve the description of the electrochemical measurements carried out: CVs in 3 electrodes configuration? What type of experiments did you carry on in a symmetric configuration? 2 electrodes charge/discharge experiments?

- Considering the chemical composition of the ternary composites obtained, this reviewer does not agree with the equations selected to carry on the corresponding calculations. What happens with the obvious pseudocapacitive contribution?

- In order to assure the stated “excellent electrochemical performance” authors should also include a comparison with previously reported values (in order to compare with similar materials)

Author Response

  1. English should be carefully reviewed.

Response: English and grammar have been carefully reviewed.

  1. In the introduction section authors compare the electrochemical performance of different aerogel-based electrode materials previously reported. However the units of the specific capacitance values are different (F/g vs. mF/cm2) in the papers selected. This issue should be considered in order to make a suitable comparative.

Response: The unit F/g is based on the weight of the active materials in the electrode, and the mF/cm2 is based on the area of the electrode. Both of them are commonly used in literatures. The calculation of the recent reported supercapacitors fabricated from cellulose are based on the area of the electrode, thus, we also use the mF/cm2 in our cases for the comparison.

  1. About contact angle measurements: how do the authors explain the results obtained? Why it has not been possible to obtain these values? What happened with the other composites?

Response: Distilled water contact angles method was run by three investigators, the obtained result are reproducible. The aim is to access the wettability of the ternary sample. As shown in Figure 1b, a water droplet penetrates into the cellulose/GO3.5/PANI sample instantly. This phenomenon could be due to the sufficient porosity on the smooth surface of the sheet. Although the contact angle was undetectable, it is satisfied enough to prove the super wettability of this cellulose-based surface, which will promote the electrolyte receptivity that is favorable for the electrochemical performance. Similar experiment have been conducted in the reference [1].

  1. FTIR experiments: authors should replace “chemical structures” by “chemical composition”. Furthermore, how do the FTIR data agree with the previously stated GO reduction during the synthesis process? The authors should be clearly clarifying this point. If the GO is reduced during the synthesis process, why it is not reflected in the corresponding spectra?

Response: “chemical structures” has been replaced by “chemical composition”. The oxygen-functional groups of GO are carbonyls (C=O), hydroxyl (-OH), carboxylic acids (CO2H), and epoxy (-O-) groups. Since the cellulose also have the hydroxyl (-OH) and the ether linkage, the reduction of oxygen-functional groups of GO can be confirmed by the dramatically reduce in the characteristic peak of carbonyls (C=O) at 1730 cm-1 in the FTIR spectrum of cellulose/GO3.5 [2]. Further, the reduction of GO also have been confirmed by Raman and XRD analyses. In the Raman spectrum of the cellulose/GO3.5, D band showed noticeably downshift to 1321 cm-1, along with an obvious increase in ID/IG of 1.47 from original 0.973. These observations suggested the thermal reduction of GO and the existence of highly disorder structures in the as-prepared cellulose/GO3.5 sample [3]. Meanwhile, the characteristic XRD peak of GO at 11° was not detected in the spectrum of cellulose/GO3.5 and cellulose/GO3.5/PANI, which also confirmed the reduction of the GO thermal blending process in the LiBr solution.

Authors state that the obtained samples showed “isotherms type IV”. Where is the hysteresis loop?

Response: The state has been revised as “All samples show type II adsorption isotherms, indicating the presence of many macropores and mesopores.”  

Electrochemical characterization:

- Authors should improve the description of the electrochemical measurements carried out: CVs in 3 electrodes configuration? What type of experiments did you carry on in a symmetric configuration? 2 electrodes charge/discharge experiments?

Response: First, to confirm the effect of GO content on the electrochemical performances of the prepared ternary composites, various cellulose/GO/PANI (prepared in 1.0 M aniline concentration) with different GO contents were initially evaluated by CV characterizations in 3-electrode configuration, as seen in Figure S2a. The results showed that the cellulose/GO3.5/PANI have the highest current response. As anticipated, the cellulose/GO3.5/PANI electrode exhibits an enhanced remarkable electrical response when compared with cellulose/GO3.5 and cellulose/PANI electrodes (Figure 6a), which can be attributed to the synergistic effect of graphene and PANI as conducting fillers [4]. Thus, we find the best doped content of GO was 3.5 wt%. Then, we characterized the GCD profiles of cellulose/GO3.5/PANI electrode in 3-electrode configuration with the comparison with that of cellulose/PANI (Figure 6c,d,e), and then we got the cyclic stability of the cellulose/GO3.5/PANI electrode (Figure 6f). Actually, to evaluate the contribution of PANI to energy storage, a series of ternary cellulose/GO/PANI samples (GO of 3.5 wt%) were prepared from different aniline concentrations of 0.5, 0.75, 1.0, and 1.25 M. Their electrochemical performances along with comparison with those of the cellulose/GO3.5/PANI electrode are presented in Figure S3. Thus, we find the ternary sample (defined as cellulose/GO3.5/PANI electrode throughout the paper) with 3.5 wt% GO and prepared in 1.0 M aniline concentration has the best capacitance performance.

Second, we assembled the symmetrical all-solid-state supercapacitors based on cellulose/GO3.5/PANI as electrodes and PVA/H2SO4 as electrolyte. The electrochemical performances of the single supercapacitor were investigated by CV, GCD, and EIS measurements, as shown in Figure 7. And the performances of the supercapacitors in parallel and in series connection are showed in Figure 8.

- Considering the chemical composition of the ternary composites obtained, this reviewer does not agree with the equations selected to carry on the corresponding calculations. What happens with the obvious pseudocapacitive contribution?

Response: The equations selected to carry on the corresponding calculations were all from the similar references [5-7], and a lot of references about the supercapacitors reported the calculations based on these equations. In our cases, the obvious pseudocapacitive contribution was from the PANI, which undergo the reversible redox reactions that redox mechanisms similar to those of rechargeable batteries [8].

- In order to assure the stated “excellent electrochemical performance” authors should also include a comparison with previously reported values (in order to compare with similar materials)

Response: The details on enhancements in the areal specific capacitance and the cyclic stability of the cellulose/GO3.5/PANI electrode are summarized in Table S1. Further, the maximum capacitive performance of our device is substantially higher than that of previously reported supercapacitors based on CNF/RGO/CNT hybrid aerogel electrode (216 mF/cm2 at 0.5 A/g) [9], PANI/Ag/CNF aerogel (176 mF/cm2 at the scan rate of 10 mV/s) [10], PPY/RGO/BC flexible electrode (790 mF/cm2 at 1.0 mA/cm2) [11], and PANI/BC/GN paper (1.93 F/cm2 at 0.25 mA/cm2) [12]. The corresponding areal energy and power density values are much higher than the recently reported supercapacitors based on CNFs/rGO/PPy (60.5μWh/cm2 at 0.1 mW/cm2) [13], CNF/RGO/CNT aerogel (28.4 μWh/cm2 at 9.5 mW/cm2) [9], PANI/CNT paper (29.4 μWh/cm2 at 391 μW/cm2)[14], RGO/MnO2 paper (35.1 μWh/cm2 at 37.5 μW/cm2) [15], rGO/PANI-PSS papers (40.7 μWh/cm2 at 500 μW/cm2) [16], PANI/PVA-H2SO4 chemical hydrogel film (42 μWh/cm2 at 160 μW/cm2) [17], CNT@PANI film (50.98 μWh/cm2 at 2294 μW/cm2) [18], CNT/activated carbon fiber felt (112 μWh/cm2 at 490 μW/cm2)[19], PANI/GN/BC paper (120 μWh/cm2 at 100 μW/cm2) [20] and carbon microfiber/MWCNT (9.8 μWh/cm2 at 189.4 μW/cm2) [21]. All this comparisons have been presented in the manuscript.

References

  1. Yin, B.-S., Zhang, S.-W., Ke, K., Wang, Z.-B., Advanced deformable all-in-one hydrogel supercapacitor based on conducting polymer: Toward integrated mechanical and capacitive performance, J. Alloys Compd. 2019, 805, 1044-1051.
  2. Zhang, J., Yang, H., Shen, G., Cheng, P., Zhang, J., Guo, S., Reduction of graphene oxide vial-ascorbic acid, Chem. Commun. 2010, 46, 1112-1114.
  3. Du, F.-P., Cao, N.-N., Zhang, Y.-F., Fu, P., Wu, Y.-G., Lin, Z.-D., Shi, R., Amini, A., Cheng, C., PEDOT:PSS/graphene quantum dots films with enhanced thermoelectric properties via strong interfacial interaction and phase separation, Sci. Rep. 2018, 8, 6441.
  4. Ge, D., Yang, L., Fan, L., Zhang, C., Xiao, X., Gogotsi, Y., Yang, S., Foldable supercapacitors from triple networks of macroporous cellulose fibers, single-walled carbon nanotubes and polyaniline nanoribbons, Nano Energy 2015, 11, 568-578.
  5. Zou, Y., Rui, L., Zhong, W., Yang, W., Mechanically robust double-crosslink network functionalized graphene/polyaniline stiff hydrogels for superior performance supercapacitors, Journal of Materials Chemistry A 2018, 10.1039.C1038TA00860D.
  6. Yu, J., Xie, F., Wu, Z., Huang, T., Wu, J., Yan, D., Huang, C., Li, L., Flexible metallic fabric supercapacitor based on graphene/polyaniline composites, Electrochimica Acta 2018, 259, 968-974.
  7. Zhou, H., Han, G., Xiao, Y., Chang, Y., Zhai, H.-J., Facile preparation of polypyrrole/graphene oxide nanocomposites with large areal capacitance using electrochemical codeposition for supercapacitors, Journal of Power Sources 2014, 263, 259-267.
  8. Eftekhari, A., Li, L., Yang, Y., Polyaniline supercapacitors, J. Power Sources 2017, 347, 86-107.
  9. Zheng, Q., Cai, Z., Ma, Z., Gong, S., Cellulose Nanofibril/Reduced Graphene Oxide/Carbon Nanotube Hybrid Aerogels for Highly Flexible and All-Solid-State Supercapacitors, ACS Appl. Mater. Interfaces 2015, 7, 3263-3271.
  10. Zhang, X., Lin, Z., Chen, B., Zhang, W., Sharma, S., Gu, W., Deng, Y., Solid-state flexible polyaniline/silver cellulose nanofibrils aerogel supercapacitors, J. Power Sources 2014, 246, 283-289.
  11. Ma, L., Liu, R., Niu, H., Zhao, M., Huang, Y., Flexible and freestanding electrode based on polypyrrole/graphene/bacterial cellulose paper for supercapacitor, Compos. Sci. Technol. 2016, 137, 87-93.
  12. Liu, R., Ma, L., Huang, S., Mei, J., Xu, J., Yuan, G., Large areal mass, flexible and freestanding polyaniline/bacterial cellulose/graphene film for high-performance supercapacitors, RSC Adv. 2016, 6, 107426-107432.
  13. Zhang, Y., Shang, Z., Shen, M., Chowdhury, S.P., Ignaszak, A., Sun, S., Ni, Y., Cellulose Nanofibers/Reduced Graphene Oxide/Polypyrrole Aerogel Electrodes for High-Capacitance Flexible All-Solid-State Supercapacitors, ACS Sustainable Chem. Eng. 2019, 7, 11175-11185.
  14. Dong, L., Liang, G., Xu, C., Ren, D., Wang, J., Pan, Z.Z., Li, B., Kang, F., Yang, Q.H., Stacking Up Layers of Polyaniline/Carbon Nanotube Network Inside Papers as Highly Flexible Electrodes with Large Areal Capacitance and Superior Rate Capability, Journal of Materials Chemistry A 2017, 5.
  15. Afriyanti, S., Ce Yao, F., Xu, W., Pooi See, L., Large areal mass, flexible and free-standing reduced graphene oxide/manganese dioxide paper for asymmetric supercapacitor device, Advanced Materials 2013, 25, 2809-2815.
  16. Chao, Y., Zhang, L., Hu, N., Zhi, Y., Hao, W., Xu, Z.J., Wang, Y., Zhang, Y., Densely-packed graphene/conducting polymer nanoparticle papers for high-volumetric-performance flexible all-solid-state supercapacitors, Applied Surface Science 2016, 379, 206-212.
  17. Wang, K., Zhang, X., Li, C., Sun, X., Meng, Q., Ma, Y., Wei, Z., Chemically Crosslinked Hydrogel Film Leads to Integrated Flexible Supercapacitors with Superior Performance, Advanced Materials 2016, 27, 7451-7457.
  18. Yu, J., Lu, W., Pei, S., Gong, K., Wang, L., Meng, L., Huang, Y., Smith, J.P., Booksh, K.S., Li, Q., Byun, J.H., Oh, Y., Yan, Y., Chou, T.W., Omnidirectionally Stretchable High-Performance Supercapacitor Based on Isotropic Buckled Carbon Nanotube Films, Acs Nano 2016, 10, 5204-5211.
  19. Dong, L., Xu, C., Yang, Q., Fang, J., Li, Y., Kang, F., High-performance compressible supercapacitors based on functionally synergic multiscale carbon composite textiles, Journal of Materials Chemistry A 2015, 3, 4729-4737.
  20. Rong, L., Ma, L., Shu, H., Jia, M., Xu, J., Yuan, G., Polyaniline/graphene/bacterial cellulose flexible electrodes for supercapacitor, New Journal of Chemistry 2016, 41.
  21. Viet Thong, L., Heetae, K., Arunabha, G., Jaesu, K., Jian, C., Quoc An, V., Duy Tho, P., Ju-Hyuck, L., Sang-Woo, K., Young Hee, L., Coaxial fiber supercapacitor using all-carbon material electrodes, Acs Nano 2013, 7, 5940-5947.

Round 2

Reviewer 3 Report

thank you so much for your kind responses to my comments. Even though I do not completely agree with the equations selected for the calculations carried out it is true that there are lots of papers using them. However, it would be highly appreciated if authors could consider (for next papers) using more appropriate equations, which also consider the pseudocapacitive contribution